# ANTH domains within CALM, HIP1R, and Sla2 recognize ubiquitin internalization signals

**Natalya Pashkova[1], Lokesh Gakhar[2,3], Liping Yu[3,4], Nicholas J Schnicker[2], Annabel Y Minard[1], Stanley Winistorfer[1], Ivan E Johnson[1], Robert C Piper[1]***

[1]Department of Molecular Physiology and Biophysics, University of Iowa, Iowa City, United States; [2]Carver College of Medicine Protein Crystallography Core, University of Iowa, Iowa City, United States; [3]Department of Biochemistry and Molecular Biology, University of Iowa, Iowa City, United States; [4]Carver College of Medicine NMR Core, University of Iowa, Iowa City, United States

**Abstract** Attachment of ubiquitin (Ub) to cell surface proteins serves as a signal for internalization via clathrin-mediated endocytosis (CME). How ubiquitinated membrane proteins engage the internalization apparatus remains unclear. The internalization apparatus contains proteins such as Epsin and Eps15, which bind Ub, potentially acting as adaptors for Ub-based internalization signals. Here, we show that additional components of the endocytic machinery including CALM, HIP1R, and Sla2 bind Ub via their N-terminal ANTH domain, a domain belonging to the superfamily of ENTH and VHS domains. Structural studies revealed that Ub binds with μM affinity to a unique C-terminal region within the ANTH domain not found in ENTH domains. Functional studies showed that combined loss of Ub-binding by ANTH-domain proteins and other Ub-binding domains within the yeast internalization apparatus caused defects in the Ub-dependent internalization of the GPCR Ste2 that was engineered to rely exclusively on Ub as an internalization signal. In contrast, these mutations had no effect on the internalization of Ste2 engineered to use an alternate Ub-independent internalization signal. These studies define new components of the internalization machinery that work collectively with Epsin and Eps15 to specify recognition of Ub as an internalization signal.

**\*For correspondence:**
robert-piper@uiowa.edu

**Competing interest:** The authors declare that no competing interests exist.

## Editor's evaluation

This manuscript describes application of x-ray crystallography and NMR spectroscopy to define how ANTH domains, which are present in endocytic machinery, bind to ubiquitin. The structural study is well-done, and results extended to the domain family by using well designed, structure-based amino acid substitutions. The Ub-binding surface was found to overlap with that used to bind VAMP8 but the authors identify mutations that retain this interaction while losing binding to Ub. These mutations are then used in yeast-based functional assays to provide new insights into Ub-dependent internalization. This study is an impressive tour de force.

## Introduction

Ubiquitination of membrane proteins dictates a variety of trafficking fates, usually ending in their degradation. Early in the biosynthetic pathway, ubiquitin (Ub) serves as a signal to mark mis-folded proteins for processing by ER-associated degradation and delivery to the Proteasome (*Lemberg and Strisovsky, 2021*; *Wu and Rapoport, 2018*). Post-ER membrane proteins undergo a variety of Ub-dependent sorting steps including transport to endosomes from the Golgi by GGAs, and MVB

sorting by ESCRTs to incorporate ubiquitinated proteins into endosomal intralumenal vesicles that are delivered to the lysosome (*Piper et al., 2014*; *Weeratunga et al., 2020*). Each of these sorting steps is mediated by an array of machinery that harbors multiple Ub-binding domains that recognize ubiquitinated proteins and, in some cases, help organize the processes they execute. At the plasma membrane, ubiquitination of membrane proteins is a signal for internalization via clathrin-mediated endocytosis (CME), a process largely conserved across eukaryotes (*Kaksonen and Roux, 2018*; *Traub and Lukacs, 2007*). Ub is a versatile signal for internalization since it can be added or removed from virtually all membrane proteins by a myriad of regulatory mechanisms (*Traub, 2009*). Studying this process in detail and evaluating the overall contribution of Ub as an internalization signal has been difficult because many proteins, including well-studied vanguard proteins that serve as model endocytic cargo, use multiple internalization signals. Furthermore, while ubiquitination of a wide variety of membrane proteins mediates their endocytosis and degradation in lysosomes/vacuoles, for many it is less clear whether Ub serves as the key internalization signal vs a sorting signal that works deeper in the endocytic pathway to convey proteins into intralumenal vesicles of multivesicular bodies (*Piper et al., 2014*). The fact that Ub can serve as an internalization signal has been clearly demonstrated with reporter membrane proteins translationally fused to Ub, which accelerates the rate of their internalization (*Traub and Lukacs, 2007*). The first instance of Ub as an internalization signal was recognized in yeast where it controls the internalization of the GPCR Ste2 that can be followed by the uptake of its ligand α-factor (*Hicke and Riezman, 1996*). Yet, Ste2 has other internalization signals that operate in tandem with Ub, likely reflecting the ability of this GPCR to be regulated at the level of internalization by multiple mechanisms (*Howard et al., 2002*). In mammalian cells, the receptor tyrosine kinase EGFR undergoes ubiquitination concomitant with its internalization. Yet EGFR also uses multiple internalization signals besides Ub to access CME and also uses different internalization routes from the plasma membrane (*Caldieri et al., 2017*; *Fortian et al., 2015*; *Goh et al., 2010*; *Pascolutti et al., 2019*; *Sigismund et al., 2013*).

A key step toward understanding the broader roles of Ub-dependent trafficking is identifying the CME machinery that recognizes Ub as an internalization signal. CME involves sequential steps of assembly and disassembly of several protein complexes including clathrin, other coat proteins, as well as adaptors that connect cargo to the internalization apparatus (*Kaksonen and Roux, 2018*; *Mettlen et al., 2018*). In the early phase, pioneering proteins localize to membrane subdomains enriched in PtdIns(4,5)P2. These in turn recruit a variety of adaptors and coat proteins concomitant with localization of cargo into the growing internalization site. The later stages of internalization are marked by the recruitment and polymerization of actin, leading to invagination and scission of the nascent endosome from the plasma membrane. Coordination between these two phases is mediated in part by proteins such as Ent1/2 and Sla2 (Epsin and HIP1R in humans) that bridge PtdIns(4,5)P2 with the actin machinery. These series of events are coordinated by multiple protein interactions that add redundancy to the system, making it more robust, while also allowing the cell to use alternate assemblies of the internalization apparatus to accommodate different cargos or respond to different regulatory pathways. Proteins that assemble in the early phase, such as AP2, have low-affinity binding sites for cargo that become operational when polymerized into coat assemblies. Within this cohort are cargo receptors for Ub. Among the best characterized candidates are Ede1/Eps15 and Ent1, Ent2/Epsin, yeast, and human orthologs, respectively, which harbor Ub-binding domains, associate with Ub-cargo, and help concentrate Ub-cargo in forming CME sites (*Traub, 2009*; *Traub and Lukacs, 2007*). Loss of these components clearly has an impact on the internalization of proteins that undergo ubiquitination as part of their internalization process. Because these proteins have multiple interactions that are collectively important for full function of the CME apparatus, their complete loss is likely to impede cargo internalization by mechanisms beyond mere loss of Ub-recognition. The strongest data for these proteins as adaptors for Ub comes from defects specifically associated with ablation of their Ub-binding domains, especially when these defects are cargo specific (*Chen et al., 2011*; *Dores et al., 2010*; *Henry et al., 2012*; *Sen et al., 2020*; *Shih et al., 2002*). Yet, the model that these components function as the sole adaptors for Ub-cargo does not represent the whole story since combined inactivation of Ub-binding domains within these proteins (eg. Ent1/2 and Ede1) in yeast does not affect Ub-dependent internalization of the Ste2 GPCR, suggesting other components help recognize Ub-internalization signals (*Dores et al., 2010*). Moreover, roles for Ub-binding domains are readily revealed when they are assessed in mutants with an endocytic apparatus weakened by the

deletion of other CME components, suggesting an alternate idea whereby Ub-binding by the endocytic machinery serves a more general function that helps organize and regulate the internalization apparatus (*Piper et al., 2014*).

Here, we find that additional components of the CME apparatus bind Ub. Ub is bound by N-terminal ANTH (AP180 N-Terminal Homology) domains from both ANTH subfamilies, which includes the CALM and AP180 subfamily as well as the subfamily including HIP1, HIP1R, and Sla2. We obtained a crystal structure of the CALM ANTH domain in a complex with Ub that we verified with NMR and mutagenesis experiments. Using this structure, we installed mutations that specifically blocked Ub-binding within the 3 ANTH-domain-containing proteins in yeast: the HIP1R homolog Sla2 and the CALM/AP180 homologs, Yap1801 and Yap1802. We followed internalization of two engineered Ste2 α-factor receptor mutants, one restricted to use only Ub as an internalization signal and another restricted to use a Ub-independent NPFxD-like motif as an internalization signal. We found that when Ub-binding was ablated within the ANTH domains in combination with other Ub-binding components of the internalization apparatus, Ub-dependent internalization was inhibited. In contrast, internalization of Ste2 that used the NPFxD internalization signal was unaffected by the loss of multiple Ub-binding sites, implying these Ub-binding sites work collectively to recognize Ub as an internalization signal of cargo.

## Results

### ANTH domains bind ubiquitin

In a previous study that used a large-scale Y2H screening method referred to as 'DEEPN', we found that the N-terminal portion of HIP1 bound to Ub (*Pashkova et al., 2016*). This Y2H interaction was observed between a Gal4-DNA-binding domain fused to two tandem copies of Ub linked via a three residue spacer and a the Gal4-transcriptional activation domain fused to residues 1–363 of mouse HIP1 (XP_036020882). This interaction was further validated by pulldown of GST-fused to residues 1–363 of mouse HIP1, which bound K63-linked polyubiquitin chains (*Pashkova et al., 2016*). This fragment contains the ANTH (AP180 N-Terminal Homology) domain and indicated that the N-terminal ANTH domain of HIP1 might be an unrecognized Ub-binding domain (*Figure 1*). GST-pulldown experiments with the ANTH domain alone from HIP1 confirmed this was sufficient for Ub-binding (*Figure 1C*).

ANTH domains are part of the larger family of ENTH (Epsin N-Terminal Homology) and VHS (Vps27, HRS, STAM) domains that are found in the N-terminus of proteins that play a variety of roles in endocytosis and many bind Ub (*Figure 1A*; *Moshkanbaryans et al., 2014*; *Takatori and Tomita, 2019*). ANTH domains are structurally similar to ENTH domains but are larger, with an additional 3 C-terminal α-helices. By sequence homology, there are two broad subgroups of ANTH domains, those found in proteins such as CALM and AP180 and those found in HIP1, HIP1R, and Sla2 (*De Craene et al., 2012*). We purified multiple recombinant ANTH domains that fell into these two subcategories (*Figure 1B*) and tested their ability to bind Ub (*Figure 1D*). Binding in solution was monitored by HSQC NMR of $^{15}$N-Ub in increasing concentrations of ANTH domains. This produced both chemical shift perturbations as well as overall peak broadening, indicating binding was in fast-exchange. Binding of Ub to $^{15}$N-labeled HIP1-ANTH was also observed in HSQC experiments, which produced selective chemical shift perturbations in backbone amides of $^{15}$N-HIP1 ANTH as well (*Figure 1—figure supplement 1*). These concentration dependent changes allowed us to derive equilibrium binding constants for each ANTH domain, ranging from 2.9 ± 0.4 μM for the mouse HIP1 ANTH domain to 33 ± 6 μM for *Drosophila* LAP, a homolog of human AP180. Mapping the largest backbone amide chemical shift perturbations in $^{15}$N-Ub (*Figure 1—figure supplement 1*) onto the structure of Ub (*Figure 1D*) indicated that all ANTH domains bound a similar surface on Ub centered on L8, I44, and V70, which mediates the vast majority of Ub-binding interactions (*Piper et al., 2014*).

### Structural basis of CALM ANTH domain bound to ubiquitin

To gain further insight into how ANTH domains bound Ub, we obtained a crystal structure of the human CALM ANTH domain in a complex with Ub resolved to 2.4 Å (*Figure 2*, *Table 1*, PDB:7JXV). These data showed that Ub bound near a loop within the C-terminal three-helical portion of the ANTH domain connecting the last two α-helices (*Figure 2A*). The structure of CALM when bound to Ub had extensive overlap with a previously determined crystal structure of CALM alone (PDB:1HFA)

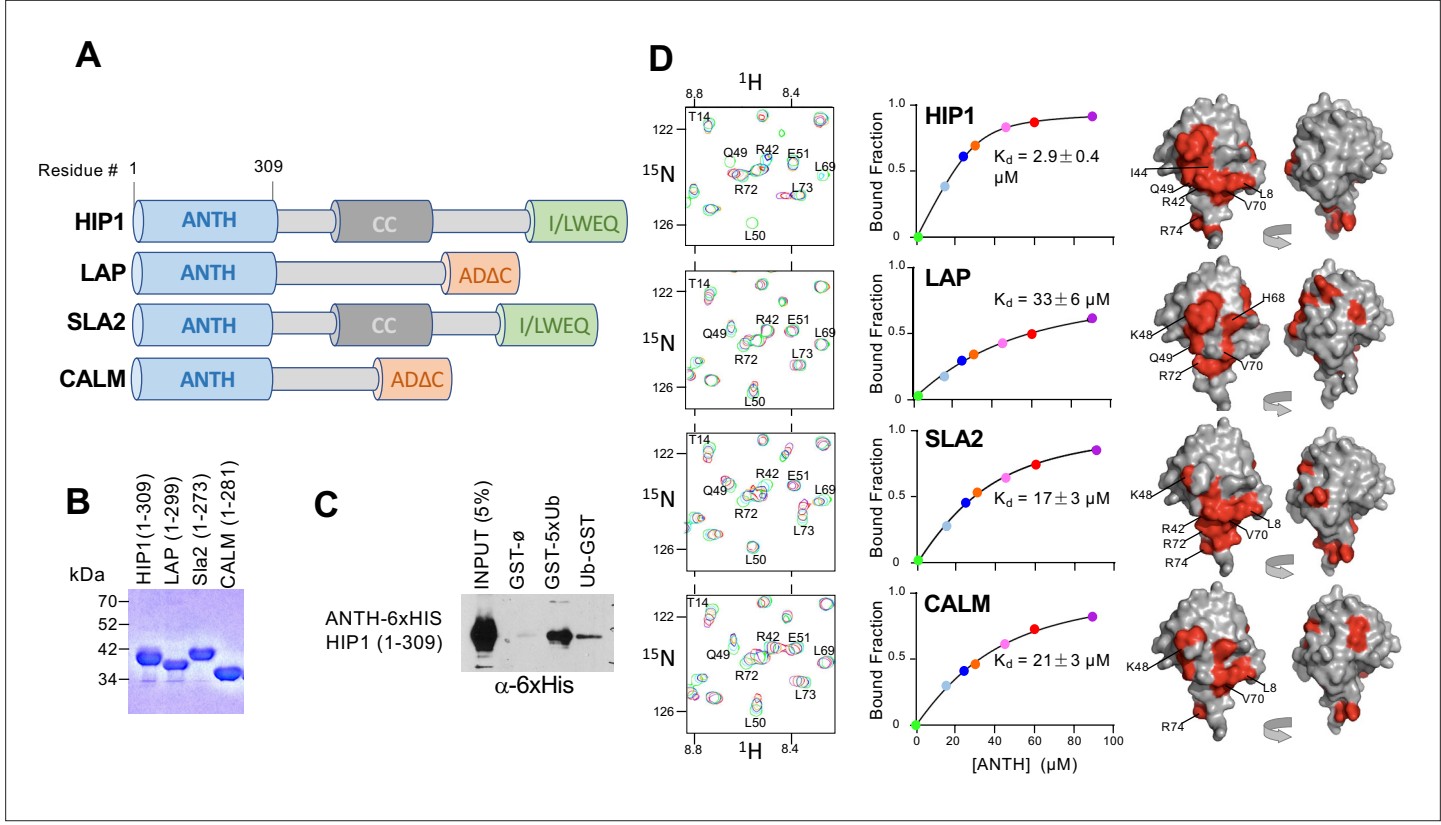

**Figure 1.** ANTH domains bind Ubiquitin. (**A**) Domain organization of proteins containing an N-terminal ANTH domain. Other domains indicated are: actin-binding I/LWEQ/THATCH (talin-HIP1/R/Slap2p actin-tethering C-terminal homology) domains; ADΔC (C-terminal Assembly Domain lacking the central clathrin and adaptor protein binding) domains (*Moshkanbaryans et al., 2014*); and CC (coiled-coil) domains that drives homodimerization (*Wilbur et al., 2008*). (**B**) SDS-PAGE and Coomassie stain of purified recombinant ANTH domains from proteins cartooned in A. (**C**) Binding of ANTH domain from mouse HIP1 using pulldown assays with GST alone (ø) or fused to the N-terminus of five linear tandem copies of Ub (GST-5xUb) or to the C-terminus of mono-Ub (Ub-GST). (**D**) Binding of [15]N-labeled Ub monitored by NMR HSQC experiments with the indicated corresponding ANTH domains in A. Titration of 30 μM [15]N-Ub in the absence (green) or presence of increasing levels of unlabeled ANTH domain. Left shows portion of HSQC spectra. Middle shows binding curves calculated from the difference in the peak intensity in absence and presence of various indicated concentrations of the ANTH domains (Kds indicated). The concentration of ANTH domains used in titration experiments are color coded to match that used in HSQC spectra. Right shows the largest chemical shift perturbations in detectable backbone resonances caused by ANTH domain binding mapped onto the molecular surface of Ub (red).

The online version of this article includes the following figure supplement(s) for figure 1:

**Figure supplement 1.** NMR data of ANTH binding Ub.

(*Ford et al., 2001*), indicating that the CALM-ANTH does not undergo large conformational rearrangements upon binding Ub (*Figure 2B*). This result is also consistent with NMR experiments with [15]N-labeled HIP1 ANTH domain showing that the HSQC spectra is largely unaltered upon Ub-binding (*Figure 1—figure supplement 1B*). Mapping the chemical shift perturbations from HSQC experiments with [15]N-Ub bound to the CALM ANTH domain (*Figure 1D* and *Figure 1—figure supplement 1B*) onto the ANTH:Ub crystal structure showed that the residues undergoing the largest changes were located at the interface captured in the crystal structure (*Figure 2C*). The CALM-ANTH:Ub interface showed hydrophobic interactions between Ub (L8, I44, V70) and CALM ANTH (F223, L274); and electrostatic interactions between Ub (R42, K48) and CALM (D224, D276) -To confirm this structure, we performed a series of Paramagnetic Relaxation Enhancement (PRE) experiments (*Figure 2—figure supplement 1A-C*) that are summarized in *Figure 2D*. PRE effects are observed in HSQC spectra when [15]N-labeled Ub residues are in close proximity to a spin-label in the CALM ANTH domain, which accelerates the relaxation of nearby [15]N[1]H spin systems and causes peak-specific broadening. We made a series of CALM ANTH domains containing single cysteine residues. Native cysteine residues were altered (C27S, C48A, and C230A) resulting in a cysteine-free ANTH domain that retained its

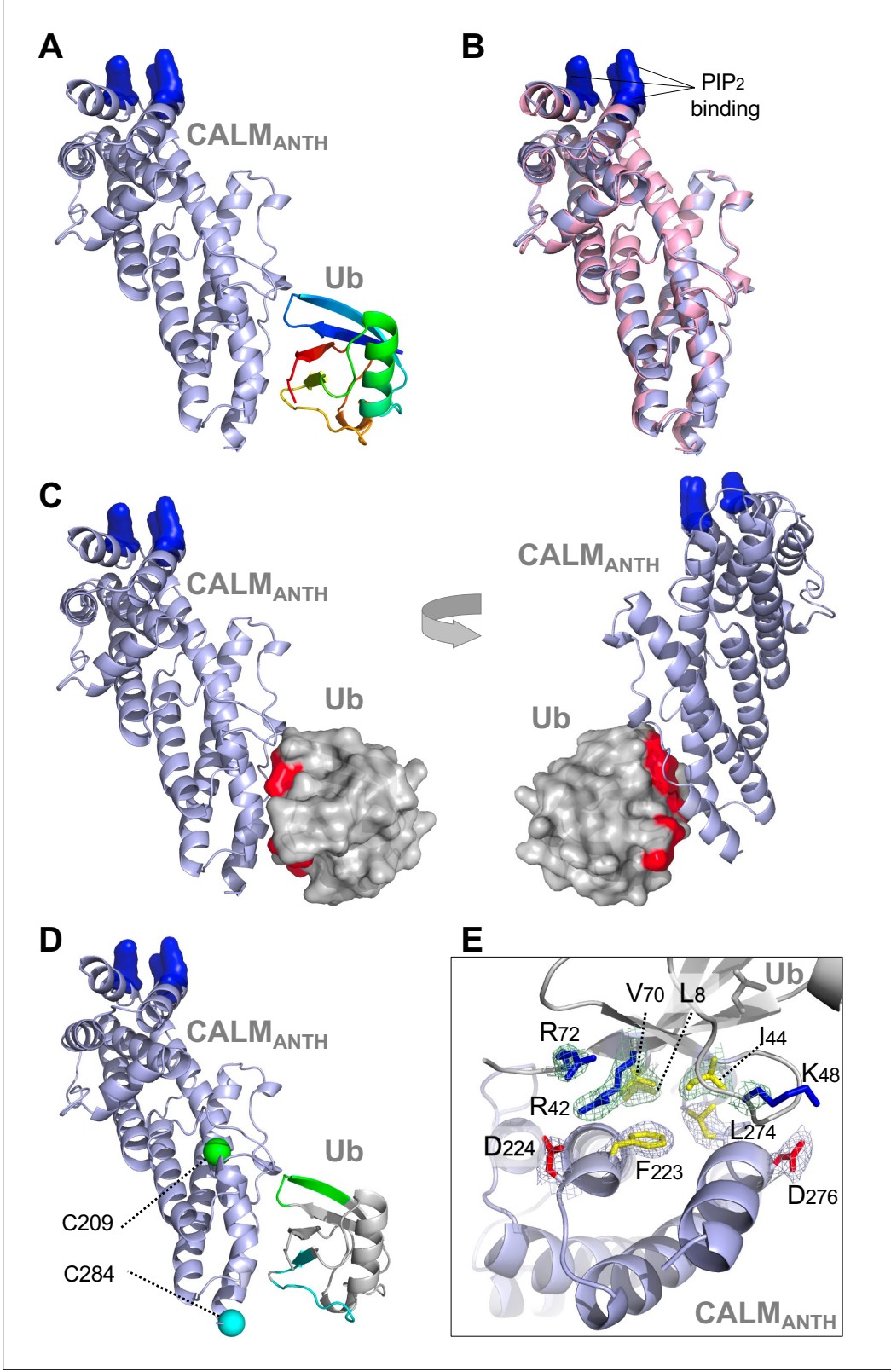

**Figure 2.** Structure of CALM ANTH domain bound to Ub. (**A**) Ribbon cartoon of the crystal structure of CALM ANTH domain (light blue) in a complex with Ub (colored from N to C terminus as blue to red (PDB:7JXV)). Basic residues in CALM that interact with PtdIns(4,5)P2 (K28, K38, K40) are highlighted in blue. (**B**) Overlay of CALM ANTH domain PDB:7JXV with CALM (residues 20–286) in the previously determined crystal structure of CALM

*Figure 2 continued on next page*

*Figure 2 continued*

alone (PDB:3ZYK, pink). (**C**) Rotated views of the CALM:Ub complex with the Ub residues undergoing strongest chemical shift perturbations in HSQC experiments when binding CALM ANTH shown in red (residues L8, R42, I44, K48, Q49, V70, R72). (**D**) Summary of paramagnetic relaxation enhancement (PRE) experiments using spin labels at either residue 209 (green sphere) or 284 (blue sphere) in CALM ANTH domain. Positions of spin labels are mapped onto the CALM:Ub structure. Residues of (30 µM) $^{15}$N-Ub undergoing the strongest PRE effect from binding (30 µM) CALM ANTH spin-labeled at 209 or 284 are colored green or blue on the ribbon cartoon. (**E**) Close-up view of residues in the ANTH:Ub interface. Electron density maps contoured to 1σ in mesh are shown for some residues (basic: blue; acidic: red; hydrophobic: yellow).

The online version of this article includes the following figure supplement(s) for figure 2:

**Figure supplement 1.** Expanded dataset of PRE experiments confirming location of Ub when bound to CALM ANTH domain.

ability to bind Ub (*Figure 2—figure supplement 1A-C*). Single cysteine substitutions were then introduced at positions 204, 209, 268, and 284, which allowed conjugation of a nitrosyl spin label, MTSL. PRE effects were measured by relative peak heights in the absence and presence of ascorbate, which reduces the nitrosyl group and inactivates the spin label. The largest PRE effects from the HSQC of $^{15}$N-Ub resulting from binding to CALM spin-labeled at residues 209, 268, and 284 were mapped onto the CALM:Ub structure and confirmed that in solution, Ub bound the same region and orientation as observed in the crystal structure.

Using the structural information above, we then made mutant versions of CALM ANTH that lacked Ub-binding (*Figure 3*). We mutated residue F223, since it sits within the hydrophobic interface with Ub, and mutated residues D224 and D276, which make electrostatic interactions with R42 and K48 of Ub (*Figure 3B*). Binding was first assessed in $^{15}$N-Ub HSQC experiments in the presence and absence of wildtype and mutant CALM ANTH domains at a 1:1 ratio. Although wildtype CALM-ANTH caused significant chemical shift perturbations within the $^{15}$N HSQC spectra, no perturbations were observed when F223 was altered to Ser or Ala alone or in combination with altering D224R or D276A (*Figure 3A*). Likewise, no Ub-binding was observed by the double mutant L274A, D276A. These results were confirmed in GST-pulldown experiments using GST fused to five tandem copies of linear Ub (*Figure 3C*) that bound the wildtype CALM ANTH domain but not a mutant ANTH domain carrying the F223A single substitution or the combined F223S and D224R double substitutions. This region of CALM is highly conserved across a subgroup of ANTH domains that includes *Drosophila melanogaster* LAP and *Saccharomyces cerevisiae* Yap1801 and Yap1802. Accordingly, mutation of residues F238 and D239 in LAP, which correspond to F223 and D224 in rat CALM, ablated detectable binding to $^{15}$N-Ub as monitored with HSQC experiments (*Figure 4*).

The CALM ANTH domain was shown previously to bind endocytic SNAREs VAMP7 and VAMP8, which is a property conserved by the *S. cerevisiae* orthologs of CALM, Yap1801, and Yap1802 (*Burston et al., 2009*; *Fujimoto et al., 2020*; *Koo et al., 2011*; *Miller et al., 2011*). To determine how these v-SNAREs bound CALM, previous studies solved a crystal structure of the CALM-ANTH domain in which the last C-terminal helix of the ANTH domain was replaced with residues 16–38 of VAMP8 (PDB:3ZYM). In this structure, the v-SNARE helix sits in the same region as the C-terminal helix of CALM (residues 261–276). This proposed binding mode for v-SNAREs substantially overlaps with that for Ub. Moreover, the binding site for Ub relies on the position of the last helical region of the ANTH domain, whereas this region would have to be dramatically repositioned to allow v-SNARE binding at the site it occupies in the crystal structure (*Figure 3—figure supplement 1A*). Thus, CALM should not be able to bind simultaneously to v-SNAREs and Ub. In view of this structure, we sought to determine whether the mutations in the CALM-ANTH that ablated Ub-binding did not adversely affect binding of v-SNAREs. This would ensure that functional analysis of the role of Ub-binding using these mutations not complicated by loss of other interactions. Therefore, we performed binding experiments using GST-VAMP7 (*Figure 3C*). Wildtype ANTH domain as well as ANTH domain containing the F223A mutation or the F223S, D224R double mutation all bound GST-VAMP7. Additionally, inclusion of excess free linear Ub (5 x) chains diminished binding of wildtype ANTH domain on GST-VAMP7 indicating that Ub and VAMP7 binding are mutually exclusive. In contrast, inclusion of 5 x linear Ub chains had no effect on the ability VAMP7 to bind to the mutant ANTH domains that lacked their Ub-binding sites. These results show that the F223A/S, D224R mutations specifically ablate Ub-binding

**Table 1.** Crystallographic data.
Data Collection and Refinement Statistics for the complex of CALM (UniProt: O55012, PICAL_RAT) and Ubiquitin (UniProt: P0CG47.1, UBB_HUMAN). Values in parathesis are for the outer shell.

| | CALM-Ub |
|---|---|
| *Data Collection* | |
| Space Group | P 4₁ 2₁ 2 |
| Unit Cell Dimensions | |
| a b c (Å) | 94.06 94.06 91.39 |
| α β γ (deg) | 90 90 90 |
| Wavelength (Å) | 1.5418 |
| Resolution Range (Å) | 18.83–2.35 (2.44–2.35) |
| Observations | 238,917 (24333) |
| Unique Reflections | 17,581 (1749) |
| Redundancy | 13.6 (13.9) |
| Completeness (%) | 99.4 (96.5) |
| I/σ (I) | 22.1 (3.2) |
| $R_{merge}$ (%) | 9.3 (90.8) |
| $R_{pim}$ (%) | 3.7 (35.9) |
| $CC_{1/2}$ | 0.999 (0.918) |
| Wilson *B*-Factor (Å²) | 40.1 |
| *Refinement* | |
| $R_{work}/R_{free}$ (%) | 20.6/26.4 |
| Number of Atoms: | |
| Protein | 2,748 |
| Water | 97 |
| Average *B*-Factor (Å²) | 48.9 |
| RMSD Bond Lengths (Å) | 0.01 |
| RMSD Bond Angles (deg) | 1.6 |
| Ramachandran favored (%) | 95.9 |
| Ramachandran allowed (%) | 4.1 |
| Clash score | 4.1 |

but not v-SNARE binding. Previous studies also identified mutations (L219S and M244K) that ablated binding of CALM to v-SNAREs. However, these mutations also abolished all detectable Ub-binding in NMR HSQC experiments with ¹⁵N-Ub (*Figure 3—figure supplement 1B*). Thus, phenotypes observed from these mutations may not solely be due to loss of v-SNARE binding.

## Ubiquitin binding to HIP1, HIP1R and Sla2 ANTH domains

We next sought mutations that disrupt Ub-binding in HIP1, HIP1R, and Sla2 ANTH domains, that belong to a different subfamily of ANTH domains (*De Craene et al., 2012*). This subfamily has limited homology with the Ub-binding region of the CALM-AP180 subfamily and residues that might coordinate Ub-binding in this ANTH subfamily were not obvious by sequence alignment alone. Secondary structure predictions and a solved crystal structure of the Sla2 ortholog from *Chaetomium thermophilum* (*Garcia-Alai et al., 2018*), provided a template to model the ANTH domains from HIP1, HIP1R, and Sla2 (*Figure 4A* and *Figure 4—figure supplement 1A-C*). These models indicated that the hydrophobic and acidic residues that reside in the loop region of the CALM ANTH domain that connect the penultimate and last α-helices where Ub binds are analogous to L212, D218 in Sla2; L250, D253 in HIP1; and L241, D244 in HIP1R. This loop region of the *C. thermophilum* Sla2 ANTH crystal structure was not resolved, likely due to flexibility, and also had the lowest level of conservation in comparison to HIP1/Sla2 family members across other species. For instance, there is no bulky hydrophobic residue at a position analogous to F223 in CALM ANTH as there is in the HIP1, HIP1R, and yeast Sla2, nor is the position of an acidic residue at the edge of the loop. We found that recombinant *C. thermophilum* Sla2 ANTH domain had negligible binding to Ub, preventing a structural analysis of Ub-binding by this ANTH domain (*Figure 4—figure supplement 2*).

We performed a series of PRE experiments to assess whether the site of Ub-binding was within the C-terminal distal portion of the yeast Sla2 ANTH domain as it is in the CALM ANTH domain. A set of Sla2 ANTH proteins each containing a single cysteine residue at different positions were spin-labeled with MTSL and used in HSQC experiments with ¹⁵N-Ub in the absence (active spin-labeled) or presence of (reduced and inactive) ascorbate to assess PRE effects (*Figure 4B*). Spin-labels placed in the N-terminal portion of the Sla2 ANTH domain did not yield PRE effects (*Figure 4B*, left) despite comparable binding overall to Ub (*Figure 4B*, middle). In contrast, spin labels in the C-terminal portion of the Sla2 ANTH domain that were near the presumptive Ub-binding site predicated by structure and sequence homology (*Figure 4—figure supplement 1A-C*) gave strong PRE effects (*Figure 4B*).

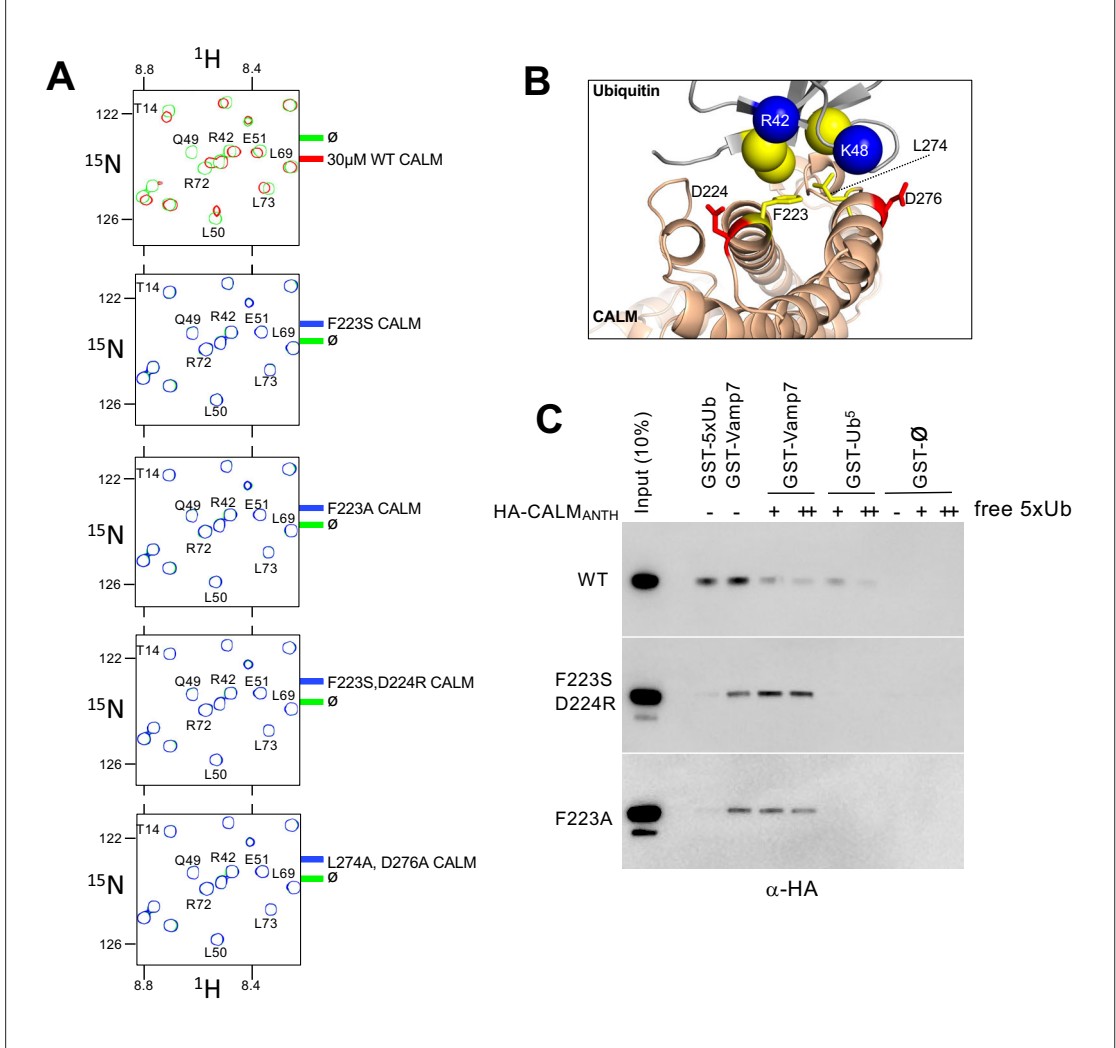

**Figure 3.** Generation of CALM lacking Ub-binding. (**A**) Region of HSQC spectra of (30 μM) $^{15}$N-Ub in the absence (green) or presence of wild type CALM ANTH domain (red) or indicated mutants (blue). Clear chemical shift perturbations of $^{15}$N-Ub are observed only in the presence of wild type CALM ANTH domain. (**B**) Cartoon of the CALM:Ub interface that when mutated abolishes Ub-binding. (**C**) GST pulldown assays of HA-epitope tagged wildtype CALM ANTH domain and the indicated mutant HA-tagged ANTH domains. ANTH domains were allowed to bind GST alone (ø), or GST fused to five linear copies of Ub or fused to Vamp7 (residues 1–188). Pulldowns were performed in the absence or presence (+:1 mg/ml; ++:5 mg/ml) of purified linear Ub (five tandem copies). Samples of Input and GSH-bead bound complexes were resolved by SDS-PAGE and immunoblotted withe α-HA antibodies.

The online version of this article includes the following figure supplement(s) for figure 3:

**Figure supplement 1.** Overlap of Ub and Vamp binding sites on CALM ANTH domain.

These data confirmed the model that Ub bound to a similar place in the Sla2 ANTH domain as it does in the CALM ANTH domain. However, without a bonafide structure of ANTH domains from human HIP1, HIP1R, or *S. cerevisiae* Sla2 we could not accurately deduce an orientation of bound Ub or a high-resolution interface. Therefore, to test our low-resolution model, we assessed the impact of mutating the bulky hydrophobic residue and acidic residues predicted in the same modeled loop region in which the Ub-binding F223 and D224 reside in the CALM ANTH domain. GST-pulldown assays showed that these residues were indeed required for efficient binding of Ub. GST fused to the HIP1 or Sla2 ANTH domain bound K63 linked polyubiquitin chains well (*Figure 4C*). Altering L250 or D253 to arginines in mouse HIP1 or L212 and D218 to alanines in Sla2 ablated detectable binding. Similarly, the human HIP1 L250S, D253R double mutant or the hHIP1R L241S, D244R double mutant failed to bind di-Ub via a yeast two-hybrid assay, in contrast to the robust interaction observed

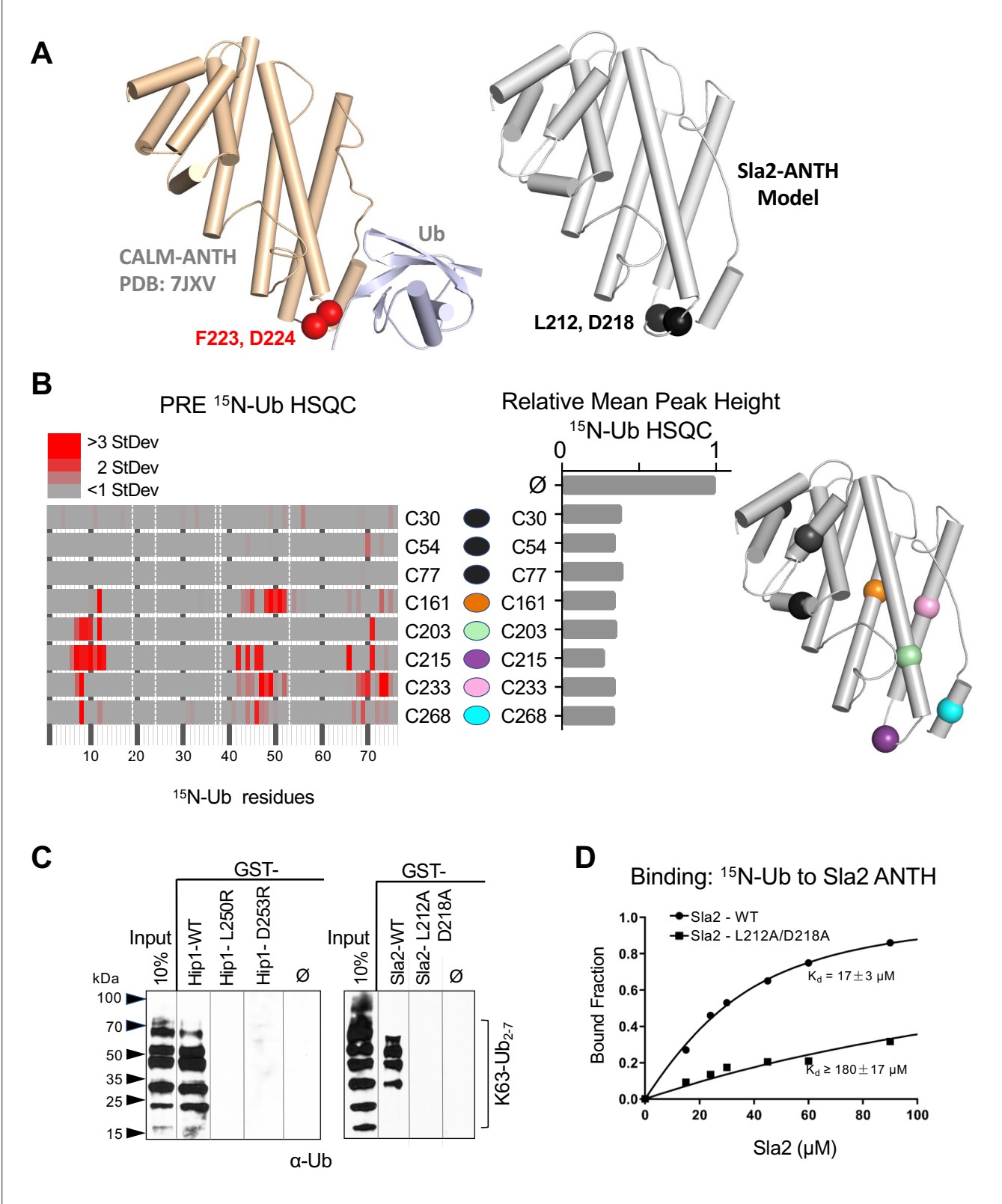

**Figure 4.** Basis of HIP1/HIP1R/Sla2 ANTH domain binding to Ub. (**A**) Cartoon representation of the CALM ANTH:Ub crystal structure (PDB: 7JXV) with helices as cylinders highlighting F223 and D224 residues required for Ub-binding (left) compared to model of *S. cerevisiae* Sla2 ANTH domain (right) generated by threading through the crystal structure of the *C. thermophilum* homolog of Sla2 (PDB:5OO7). Structurally analogous residues (L212 and D218) are shown in black spheres. (**B**) PRE effect on Ub by Sla2 binding. Residues undergoing detectable PRE are plotted by residue number. A set of

*Figure 4 continued on next page*

*Figure 4 continued*

Sla2 ANTH domains with single residues that were spin-labeled with MTSL and used in a series of PRE experiments with $^{15}$N-Ub. Left shows detected PRE effects over 1 StDev for the dataset color coded by residue number. Residues absent in the HSQC spectra are designated by white dashed lines. Middle shows relative average peak heights of all HSQC backbone amide peaks of (30 μM) $^{15}$N-Ub in the absence (ø) or presence of reduced MTSL-labeled Sla2 ANTH (in reduced state) at the indicated positions (30 μM). Right, positions of the MTSL-labels mapped onto the model of Sla2 ANTH domain. (**C**) GST pulldown assays using GST alone (ø) or fused to wildtype mouse HIP1 or *S. cerevisiae* Sla2 ANTH domains or the indicated mutants. A mixture of polyubiquitin chains (K63-linked) of different lengths were allowed to bind. Samples of input and GSH-bead-bound complexes were resolved by SDS-PAGE and immunoblotted with α-Ub antibodies. (**D**) NMR titrations using 30 μM $^{15}$N-Ub in the presence of increasing concentrations of mutant (L212A, D218A) or wildtype Sla2 ANTH domain. Binding data were analyzed using selective amides detected in the HSQC spectra.

The online version of this article includes the following figure supplement(s) for figure 4:

**Figure supplement 1.** Analysis of HIP1/Sla2/HIP1R Ub-binding.

**Figure supplement 2.** Impact of mutations in Sla2 and LAP on Ub-binding.

with wildtype human HIP1 and human HIP1R ANTH domains (*Figure 4—figure supplement 1D, E*). NMR titrations with mutant L212A, D218A Sla2 ANTH domain showed a > 10 fold loss of apparent Kd for Ub-binding in comparison with wildtype Sla2 ANTH domain (*Figure 4D*). Even less binding was observed in HSQC experiments when Sla2 ANTH carried an L212S, D218R double mutation (*Figure 4—figure supplement 2*). This placement of Ub-binding coincides well with the position of the CALM Ub-binding site, which is well conserved in this subfamily of ANTH domain proteins in yeast and *Drosophila* (*Figure 4—figure supplement 1F, G*). Together our mutational studies based on structure predictions combined with PRE experiments that verified the proximity of Ub-binding verified a similar binding site in the HIP1/HIP1R/Sla2 subgroup of ANTH domains that is observed in the CALM/LAP/AP180 subgroup of ANTH domains.

## Cargo-specific role for ub-binding by ANTH domains

We next examined the role of Ub-binding by ANTH-domains in internalization from the cell surface (*Figure 5*). Many previous studies have found that a variety of components of the internalization machinery possess Ub-binding domains (*Figure 5B*, and *Figure 5—figure supplement 1A*), though it is not clear whether these might be functionally confined to only binding Ub-cargo, or fulfill some generalized regulatory function for the internalization process, or be dedicated to another function altogether (*Traub and Lukacs, 2007*). We pursued this question in yeast, where many aspects of CME are similar to that of animal cells and offer the capacity of combining systematic mutations in multiple proteins expressed from their endogenous loci (*Kaksonen and Roux, 2018*; *Mettlen et al., 2018*).

Internalization in yeast can be followed by ligand-stimulated internalization of the α-factor receptor, a GPCR that undergoes ligand-dependent phosphorylation that then allows its ubiquitination and endocytosis (*Hicke et al., 1998*; *Rohrer et al., 1993*). Previous studies have shown that Ste2 can use Ub as an internalization signal and/or NPFxD-like sequence, the latter of which binds to an Sla1-homology domain in Sla1 (*Howard et al., 2002*). Ste2 lacking its NPFxD motif, made by truncation at residue 365, relies on ubiquitination for an internalization signal. Whereas Ste2 lacking ubiqitinatable lysines in its C-terminal tail relies on Ub-independent/NPFxD-mediated internalization via Sla1 binding (*Figure 5A*). We reasoned that if Ub-binding ANTH domains served as specific adaptors to internalize ubiquitinated cell surface proteins, loss of Ub-binding would cause defects in Ub-dependent internalization of Ste2 without affecting NPFxD-dependent internalization of Ste2. If instead, Ub-binding served a general regulatory role for the internalization apparatus, loss of Ub-binding would impose defects on both Ub-dependent and Ub-independent routes of Ste2 internalization.

To measure internalization, we followed the subcellular distribution of a synthetic α-factor peptide that was conjugated to rhodamine (α-factor$^{RhD}$). Previous experiments indicated that the central lysine (K7) of α-factor likely points away from the ligand binding site in Ste2, allowing it to carry a fluorescent dye without loss of activity (*Naider and Becker, 2020*). In addition, norleucine was substituted for Met12 in α-factor, making it resistant to inactivation by oxidation. Cells kept at 0 °C were labeled with α-factor$^{RhD}$, washed in cold buffer and warmed while imaged by epifluorescence microscopy (*Figure 5C*). Deconvolved images were measured for the amount of fluorescence remaining at the cell surface and data were fit to a single exponential decay. As expected, internalization of wildtype Ste2 was rapid and complete within 10 min (*Figure 5D* and *Figure 5—figure supplement 1B, C*). In contrast, truncated Ste2 lacking its NPFxD-like motif yet

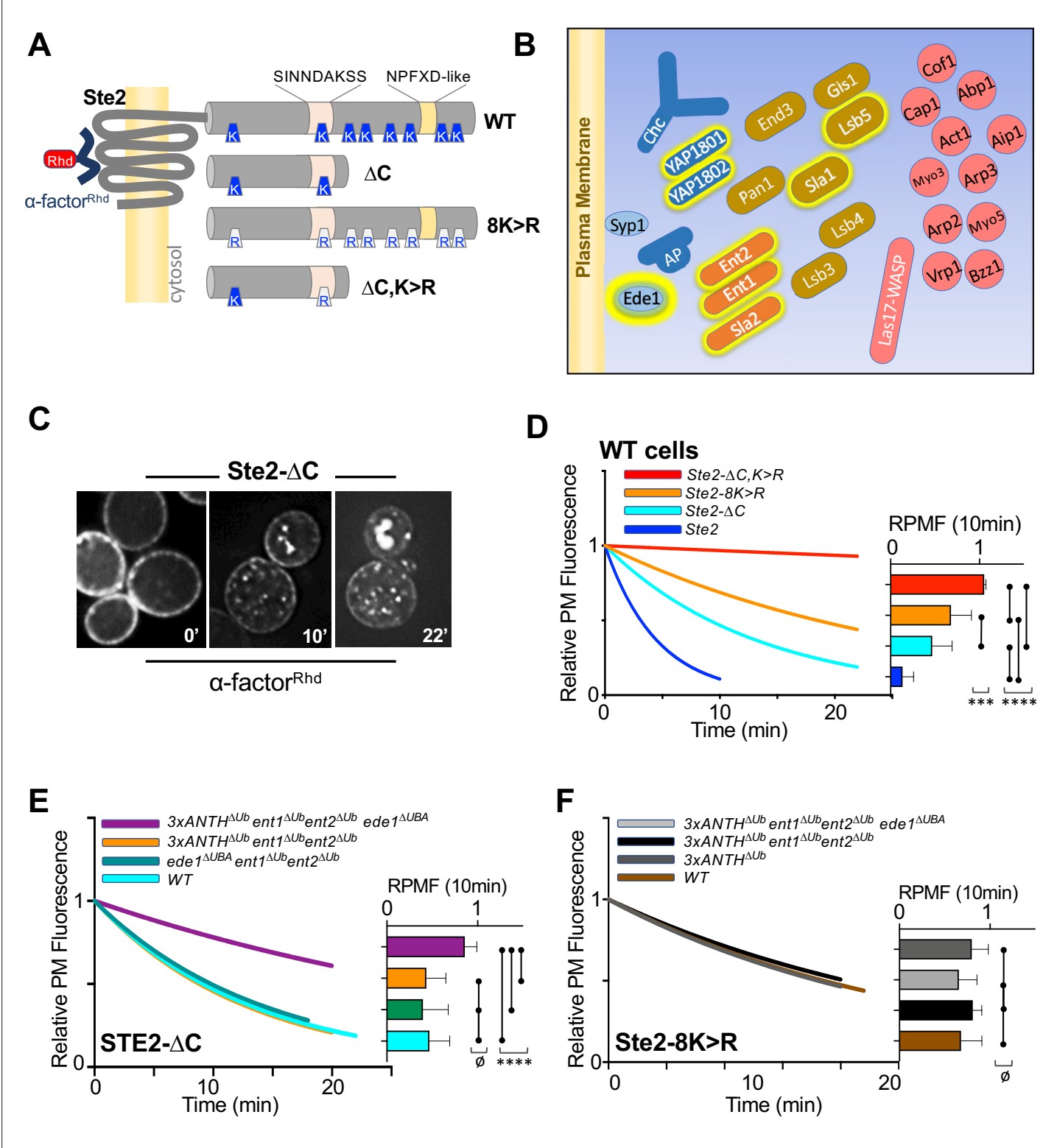

**Figure 5.** Role of ANTH domains on Ub-dependent internalization. (**A**) Schematic of Ste2. Full-length Ste2 (upper) has redundant internalization signals: lysine (**K**) residues serve as acceptors for ubiquitination and an NPFXD-like motif (residues 392–396) that allows for Ub-independent internalization. Also shown is the serine-containing region (residues S-331-INNDAKSS) that undergoes ligand dependent phosphorylation that is required for ligand-stimulated internalization. (**B**) Schematic of cellular machinery that catalyzes clathrin-associated internalization in yeast, with proteins arriving earliest to the site of internalization placed closer to the plasma membrane. Proteins highlighted with a yellow border are those identified in this and previous

*Figure 5 continued on next page*

*Figure 5 continued*

studies as harboring Ub-binding domains. (**C**) Internalization assays following localization of Rhodamine(RhD)-labeled α-factor bound to the surface of cells at 0 °C for 1h, washed and then allowed to warm to 20 °C during imaging. Sample micrographs of deconvoluted images of yeast expressing Ste2-ΔC allowed to endocytose α-factor$^{Rhd}$ for 0, 10, and 22 min at 20 °C. (**D**) Quantitative assays were performed in wildtype yeast strains expressing the indicated forms of Ste2. Image stacks of cells throughout the time course were collected, deconvolved, and quantified for the percentage of fluorescence at the cell surface. Data were fit to a one-phase exponential decay and normalized to the 0 time point. Insert at right shows the mean ± StDev level of normalized α-factor$^{Rhd}$ at the plasma membrane 10 min after warming. Statistical significance of indicated pairs were calculated using one-way ANOVA for all conditions (p: ≥ 0.05 = ø, ≤ 0.05 = *, ≤ 0.01 = **, ≤ 0.001 = ***, ≤ 0.0001 = ****) (**E**) Internalization of α-factor$^{Rhd}$ by Ste2-ΔC in yeast strains carrying the indicated mutations. Fitted lines to data normalized to the 0 time point. The proportion of surface-localized α-factor$^{Rhd}$ at 10 min (mean ± StDev) is also shown. (**F**) Internalization of α-factor$^{Rhd}$ by Ste2-8K > R in yeast strains carrying the indicated mutations.

The online version of this article includes the following figure supplement(s) for figure 5:

**Figure supplement 1.** Expanded dataset for Ub-dependent internalization of α-factor.

**Figure supplement 2.** Method for quantifying images of α-factor$^{Rhd}$ distribution.

still possessing ubiquitinatable lysines (Ste2-ΔC), had an intermediate rate of internalization. Likewise, Ste2 containing an intact NPFxD motif but carrying K > R mutations (Ste2-8K > R) within its cytosolic domain also internalized at an intermediate rate, though somewhat slower than the truncated Ste2. In contrast, a truncated mutant Ste2 lacking its NPFxD motif and its C-terminal ubiquitinatable lysines (Ste2-ΔC,8K > R) showed almost no internalization. Expression of Ste2 variants was done from the *STE2* loci rather than low-copy plasmids, which produced high variability in the level of surface labeling with α-factor$^{RhD}$.

We assessed strains which carried mutations in their endogenous loci that inactivated Ub-binding in Sla2 (Sla2$^{ΔUb}$: L212A, D218A), Yap1801 (Yap1801$^{ΔUb}$: F213A, E214S) and Yap1802 (Yap1802$^{ΔUb}$: F207A, E208A). When all these mutations were combined (3xANTH$^{ΔUb}$), internalization assays showed that α-factor$^{RhD}$ uptake by Ste2 via its Ub-dependent (Ste2-ΔC) (*Figure 5E* and *Figure 5—figure supplement 1B-C*) or Ub-independent routes (Ste2-8K > R) (*Figure 5F* and *Figure 5—figure supplement 1B, C*) was unaffected. We then combined these mutations with ones in other Ub-binding proteins that participate in internalization. Previous studies showed that Ent1 and Ent2 each possess 2 UIM (Ub-Interaction Motifs) whereas Ede1 contains a C-terminal UBA (UB-Associated) domain. Combined loss of Ub-binding activity of Ent1/2 and Ede1 had no effect on α-factor$^{RhD}$ internalization (*Figure 5E*) in agreement with previous studies (*Dores et al., 2010*). Likewise, loss of Ub-binding by Ede1 or loss of Ub-binding in both Ent1 and Ent2 had no affect on internalization when each was combined with loss of Ub-binding in all ANTH domains. In contrast, marked reduction in α-factor$^{RhD}$ internalization by Ste2ΔC was observed in strains carrying all of these mutations in which Ub-binding mutations in Ent1/2 and Ede1 were combined with Ub-binding mutations in all of the ANTH domain proteins, Sla2, Yap1801, and Yap1802 (*Figure 5E* and *Figure 5—figure supplement 1C*). Thus, all the Ub-binding domains we evaluated appear to contribute redundantly to internalization of Ste2ΔC, which uses ubiquitination as its internalization signal. We also assessed the effect of these mutant combinations on the internalization of Ste2-8K > R, which has an NPFxD interanalization motif. Here, combined loss of Ub-binding by the three ANTH domains in addition to Ent1/2 and Ede1 had no effect on α-factor$^{RhD}$ internalization (*Figure 5F*). Thus, the combination of mutations that inactivated Ub-binding by a whole cohort of Ub-binding proteins was selective for a cargo that uses Ub as an internalization signal, demonstrating that the general ability of cells to execute and regulate internalization was largely intact.

We did note that while internalization of Ste2-ΔC in the combined 3xANTH$^{ΔUb}$, Ent1/2$^{ΔUb}$, Ede1$^{ΔUb}$ mutant was greatly slowed (*Figure 5E*), it was not reduced to the level of Ste2-ΔC,K > R lacking Ub and NPFxD internalization signals altogether (*Figure 5D*). One possibility is that the mutations we used to inactivate Ub-binding by Sla2 ANTH (Sla2$^{ΔUb}$: L212A,D218A), which diminished the Kd of Ub-binding by 10 fold (*Figure 4D*), were not severe enough to ablate all functional Ub-binding. Thus, we also analyzed an alternative mutant of Sla2 ANTH (Sla2$^{*ΔUb}$: L212S,D218R) with even weaker binding to Ub (*Figure 4—figure supplement 2*). However, in the context of combined mutations in the Ub-binding domains of Yap1801, Yap1802, Ent1/2 and Ede1, the Sla2$^{*ΔUb}$ mutant (L212S, D218R) did not reduce internalization beyond what was observed for the Sla2$^{ΔUb}$ (L212A,D218A) mutant in the same context (e.g. 3xANTH$^{ΔUb}$, Ent1/2$^{ΔUb}$, Ede1$^{ΔUb}$) (*Figure 5—figure supplement 1*).

## Discussion

Our data show that ANTH domains bind Ub and that this plays a role in endocytosis. Our functional data show that a set of Ub-binding domains found within several components of the endocytic apparatus can work together to recognize and internalize ubiquitinated cell surface proteins.

The three ANTH domain-containing proteins in yeast are Sla2 (the ortholog of human HIP1 and HIP1R) and Yap1801 and Yap1802 (orthologs of human AP180 and CALM). Sla2 plays a major role in internalization in yeast and its loss causes severe defects in the internalization of proteins and lipid dyes and disrupts the ordered assembly of the endocytic apparatus and transition to actin polymerization (*Brach et al., 2014*; *Kaksonen et al., 2003*; *Newpher et al., 2005*; *Skruzny et al., 2015*; *Wesp et al., 1997*). Sla2 forms a complex with Ent1/2 (yeast Epsins) that can bridge PtdIns(4,5)P2-containing membranes with the actin cytoskeleton via its PtdIns(4,5)P2-binding ANTH domain and C-terminal THATCH domains, respectively (*Skruzny et al., 2015*). Sla2 arrives early in the assembly process and remains during actin polymerization making it a plausible candidate cargo receptor. Yap1801 and Yap1802 play a more nuanced role in internalization. Both localize with clathrin at the cell surface and arrive in the early phase of CME assembly (*Carroll et al., 2012*). Loss of both has no effect on GPCR internalization but does cause accumulation of the v-SNARE Snc1 at the cell surface implying they work as dedicated cargo-specific adaptors for a subset of proteins (*Burston et al., 2009*; *Huang et al., 1999*; *Maldonado-Báez et al., 2008*). More severe phenotypes are revealed when *yap1801/2Δ* double deletion mutations are combined with other mutations in the endocytic apparatus suggesting that these proteins may provide additional functions made redundant in the context in wild type cells with a full complement of endocytic machinery (*Dores et al., 2010*; *Maldonado-Báez et al., 2008*). Nonetheless, these earlier results suggested that YAP1801/2 play some role in supporting the ability of the CME apparatus to mediate Ub-dependent internalization.

Our studies, that combined precise mutations within genes at their endogenous loci while also following internalization of a cargo protein constrained to rely on Ub as an internalization signal, reveal a role of ANTH-domain proteins receptors/adaptors for ubiquitinated proteins. An alternative idea for the role of Ub-binding within the internalization apparatus is that it is used to control the assembly and activity of the CME machinery. This idea is supported by experiments that show general defects in endocytosis resulting from deleting Ub-domains in combination with loss of other genes and by the observation that the endocytic machinery also can undergo ubiquitination itself (*Dores et al., 2010*; *Maldonado-Báez et al., 2008*; *Polo et al., 2002*; *Savio et al., 2016*; *Sugiyama et al., 2005*; *Tanner et al., 2019*; *Weinberg and Drubin, 2014*; *Woelk et al., 2006*). In some ways, these ideas may be a false dichotomy in view of studies that show cargo itself can influence the assembly process, providing a way that recognition of Ub as an internalization signal is the mode through which Ub-binding domains control assembly (*Carroll et al., 2012*; *Henry et al., 2012*; *Loerke et al., 2009*; *Maib et al., 2018*; *Pedersen et al., 2020*; *Puthenveedu and von Zastrow, 2006*). The presence of Ub-cargo might help form endocytic vesicles of different constituents that can determine different fates (*Sigismund et al., 2021*), or drive more processive assembly of clathrin-coated vesicles by satisfying cargo checkpoints that Ub-binding components control (*Chen and Schmid, 2020*). Our data help distinguish a clear role for Ub-binding as a way to recognize Ub as a discrete internalization signal. This conclusion rests on the observation that eliminating Ub-binding sites in ANTH domains in addition to other CME components specifically inhibits internalization of Ub-cargo while leaving endocytosis of proteins that rely on other internalization motifs unperturbed. The functional effect we tested was at the level of cargo internalization. Yet, there may well be many alterations in the assembly kinetics of the endocytic apparatus (*Kaksonen and Roux, 2018*; *Mettlen et al., 2018*) that result from altering multiple Ub-interactions which are not apparent when singly eliminated (*Lu and Drubin, 2017*). Such alterations might not dramatically affect the overall internalization of cell surface proteins since yeast cells lacking multiple components of the endocytic apparatus display measurable changes in the assembly of the actin-based internalization machinery yet internalize proteins and lipophilic dyes (e.g. FM4-64) relatively well (*Brach et al., 2014*; *Prosser et al., 2011*). Thus, a deeper analysis of how the assembly process could be altered in these combinatorial mutants will be important to assess other roles for Ub-binding. The identification of additional Ub-binding sites within the CME apparatus coupled with a variety of new tools for locally perturbing ubiquitination (*Huang et al., 2013*; *Stringer and Piper, 2011*; *Wu et al., 2020*) as well as a wealth of high-resolution approaches that examine the assembly and architecture of CME (*Kaksonen and Roux, 2018*; *Mettlen et al., 2018*) can enable

those investigations. In addition, the ability to better perturb endocytosis of proteins that use Ub as an internalization signal provides a way to determine how prevalent and biologically important the use of Ub as internalization signal actually is. For instance, one aspect of CALM function is that it is correlated with Alzheimer's Disease in GWAS studies (*Harold et al., 2009*). Perturbing CALM experimentally alters the trafficking of proteins associated with driving Alzheimer's Disease pathogenesis and it will be of interest to determine whether some of those effects are due to the function of CALM as an adaptor for ubiquitinated proteins (*Harel et al., 2011*; *Kanatsu et al., 2016*; *Kanatsu et al., 2014*; *Moreau et al., 2014*; *Wu et al., 2009*).

In mammalian cells, the Ub-binding Eps15 and especially Epsin1 proteins appear to contribute to recognition of Ub as an internalization signal, with the most compelling experiments involving specific defects in the internalization process that can be correlated with specifically altering Ub-binding domains (*Barriere et al., 2006*; *Bertelsen et al., 2011*; *Chen et al., 2011*; *Fortian et al., 2015*; *Gucwa and Brown, 2014*; *Hawryluk et al., 2006*; *Kazazic et al., 2009*; *Szymanska et al., 2016*; *Vuong et al., 2013*; *Wang et al., 2006*). The ability to follow cargo internalization vs post-internalization processes such as delivery to the vacuole or decreased recycling is more challenging in yeast. However, endocytosis of one Ub-cargo in yeast is particularly reliant on Ub-binding by Epsin (Ent1/2), further supporting the idea it might work as a Ub-dependent adaptor (*Sen et al., 2020*). For other yeast cargo such as Ste2, removing Ub-binding from Ede1 and Ent1/2 together has no effect on internalization (*Dores et al., 2010*), unless also combined with eliminating other Ub-binding sites in ANTH domains as shown here. Future work is required to find how mammalian ANTH-domain containing proteins cooperate with Eps15 or Epsin to internalize Ub-cargo and whether that balance is universal or tissue and cargo specific. A possibility emerges that the combination of ENTH and ANTH domain proteins converge to form the core Ub-recognition machinery for CME. Indeed, Ent1/2 and Sla2 collaborate to form a coated structure complex catalyzed around PtdIns(4,5)P2 (*Garcia-Alai et al., 2018*; *Lizarrondo et al., 2021*; *Skruzny et al., 2015*). Although there are differences between the respective yeast and human orthologs, Epsin and Hip1R form oligomers as well (*Garcia-Alai et al., 2018*; *Messa et al., 2014*). In addition, although the other subfamily of ANTH domains in proteins such as AP180 and CALM do not readily form oligomers in vitro with Epsin/Ent1,2, they have similar assembly kinetics and occupy the same region within forming clathrin-coated vesicles as Epsin/Hip1R determined by CLEM and FRET studies (*Brach et al., 2014*; *Skruzny et al., 2020*; *Sochacki et al., 2017*).

Despite our ability to specifically slow Ub-mediated internalization of a cargo that exclusively relies on Ub as an internalization signal, the residual rate was still higher than that of cargo stripped of its ubiquitinatable lysine residues. This implies there are additional Ub-receptors. We found no evidence that the Ub-binding component Lsb5 (*Costa et al., 2005*) fulfilled that role (*Figure 5—figure supplement 1C*) leaving the most likely candidate to be Sla1, which harbors an SH3 domain that binds Ub (*He et al., 2007*; *Stamenova et al., 2007*). Sla1 interacts with Sla2 and serves as the receptor for NPFxD internalization motifs, which operates in full-length Ste2, so it is positioned effectively in the CME assembly cascade to also work as a *bonafide* Ub-adaptor (*Gourlay et al., 2003*; *Howard et al., 2002*; *Sun et al., 2015*). One of the issues that arises from this study is why there are many Ub-binding domains. Other cellular machines that process ubiquitinated proteins also possess multiple Ub-binding sites, including the proteasome, which degrades ubiquitinated cytosolic proteins (*Sahu and Glickman, 2021*), and the ESCRT machinery, that sorts ubiquitinated proteins into endosomal intralumenal vesicles (*Piper et al., 2014*). Many of the same hypotheses to explain multiple Ub-binding interfaces in these complexes are also plausible for the internalization apparatus. One possibility is that different types of Ub-cargo are optimally captured by different Ub-binding components. These differences could be due to where the Ub moiety is in proximity to the membrane or be due to the type of Ub linkage on the cargo. Although mono-Ub as well as different poly-Ub topologies can serve as internalization signals (*Boname et al., 2010*; *Goto et al., 2010*; *Lauwers et al., 2010*; *Okiyoneda et al., 2010*; *Traub and Lukacs, 2007*), each might require specific Ub-binding components to sort cargo efficiently into forming CCVs (Clathrin-coated vesicles). Where Ub is attached onto cargo might also play a role. ANTH domains orient with their N-termini near the membrane, by binding PtdIns(4,5)P2 (*Ford et al., 2001*; *Miller et al., 2011*; *Sun et al., 2005*) as well as binding either similarly tethered Ent1 or Epsin (*Garcia-Alai et al., 2018*; *Itoh et al., 2001*; *Lizarrondo et al., 2021*; *Messa et al., 2014*; *Skruzny et al., 2015*) or inserting an N-terminal helix into the lipid bilayer (*Miller et al., 2015*). This

would serve as a mechanism to space the Ub-binding domains at a set distance from the membrane and be contained within the clathrin lattice where they would be constrained to interact with Ub near the membrane. In contrast, other Ub-binding domains in Ent1/2 and Epsin or Ub-binding sites in Ede1 and Eps15 (*Figure 5—figure supplement 1A*) lie within long flexible regions that could sample Ub attached at a number of positions on the cytosolic regions of cargo. Another possibility is that the ability of these proteins to work as adaptors to bring Ub-cargo into an internalization site may actually reflect a secondary function revealed experimentally when other mechanisms for Ub-cargo sorting are eliminated. That is, these Ub-binding proteins might serve as cargo sensors, recruited or stabilized by Ub-cargo to allow CCVs to be programed with a different cohort of adaptors that determine different internalization routes (*Cao et al., 1998*; *Fortian et al., 2015*; *Mundell et al., 2006*; *Pascolutti et al., 2019*), recruit additional proteins that work downstream and direct different endocytic fates of their cargo (*Gucwa and Brown, 2014*; *Manna et al., 2015*; *Mayers et al., 2013*; *Rai et al., 2014*; *Roxrud et al., 2008*; *Sigismund et al., 2008*), or to alter assembly dynamics to hasten CCV completion and extinguish signaling which might otherwise be enhanced by extended clathrin-coated pit lifetimes (*Eichel et al., 2016*; *Flores-Otero et al., 2014*). Finally, these Ub-binding proteins may be uniquely important for roles outside the internalization apparatus, such as transcription, autophagy and endosomal function (*Borlido et al., 2009*; *Brodsky et al., 2014*; *Moreau et al., 2014*; *Tian et al., 2013*; *Tsushima et al., 2013*; *Wilfling et al., 2020*).

## Materials and methods

### Reagents

Plasmids used for protein expression are described in *Supplementary file 1*. *E. coli* bacteria were grown in LB media or 2xYT media. Yeast strains were grown in standard synthetic media (SD) or rich media (YPD). Yeast strains used in this study are described in *Supplementary file 2*. Plasmids were made by Gibson assembly cloning (NEB BioLabs, Ipswich, MA). Synthetic DNA fragments were obtained from IDT (Coralville, IA). Plasmid point mutations were generated using *Pfu Ultra* polymerase by Quick Change PCR mutagenesis (Agilent Technologies, Santa Clara, CA). The resulting DNA constructs and strains were verified by DNA sequencing.

The antibodies used were: monoclonal anti-His THE HIS Tag (GenScript, Piscataway, NJ); monoclonal anti-HA (BioLegend, San Diego, CA); monoclonal anti-Ub P4D1 (Santa Cruz Biotechnology Inc, Santa Cruz, CA); polyclonal anti-c-Myc (QED Bioscience Inc, San Diego, CA). The α-factor[Rhd] peptide was custom synthesized by LifeTein LLC, Hillsborough, NJ: W-H-W-L-Q-L-K-(Ahx-TAMRA)-P-G-Q-P-Nle-Y where Nle is norleucine, Ahx is aminohexanoic acid and TAMRA is tetramethyl Rhodamine. The peptide was resuspended in DMSO as 10 mg/ml stock kept at –20 °C. For NMR PRE experiments, MTSL [S-(1-oxyl-2,2,5,5-tetramethyl-2,5-dihydro-1H-pyrrol-3-yl)methyl methanesulfonothioate] was used as a niroxide spin label of proteins with single Cysteine (TRC Inc, Toronto, Canada).

### Fluorescence microscopy and α-factor[Rhd] internalization

We initially used Ste2 fused to the pH-sensitive GFP, pHluorin, which could be extinguished upon delivery to the slightly acidic vacuole (*Miesenböck et al., 1998*; *Prosser et al., 2016*). However, our preliminary experiments using dual-labeling with the addition of α-factor[RhD] indicated that while pHluorin did not confer an acceptor site that could promote internalization, the presence of pHluorin slowed uptake of α-factor[RhD] by Ste3-ΔC and the conditions we used did not fully quench fluorescence in the vacuole, prompting us to use untagged Ste2 mutants (*Figure 5—figure supplement 1A, B*).

Images were acquired using Olympus fluorescent microscope BX60 controlled by iVision software (BioVision Technologies) and equipped with UPlanSApo100×/1.40 oil objective and Hamamatsu Orca-R2 digital camera (Hamamatsu, JP).

To define plasma membrane localization, yeast strains were expressing pFur4-GFP missing 60 a.a. at the N-terminus and thus failed to internalize (*Marchal et al., 2000*). Cells were grown overnight at 29 °C in liquid synthetic SD-Ura-Met media to $OD_{600}$ = 0.2–0.4, spun down (1–1.5 ml) and resuspended in 300–500 µl SD media pH = 6.7.

The aliquot of 50 µl cells was transferred in small glass tube pre-chilled on ice with added 5 µl BSA (10 mg/ml stock) and 1 µl α-factor[Rhd] peptide (1 mg/ml sock in $H_2O$), covered from light and incubated 1 hr on ice mixing occasionally. Labeled cells were spin-washed at 2 °C in Eppendorf tubes with

ice-cold SD media 3 × 1 ml then resuspended in 20 μl SD media. Labeled cells kept on ice protected from the light up to 2 hr while 2 μl aliquots were taken and applied on a pre-chilled glass slide to initiate a microscopy time series. Images were taken at 20 °C with the time zero delayed by 30 s to allow cells to warm and to focus on cells. For each timepoint, the field of view was changed to prevent photobleaching. Five-slice images stacks with GFP and Rhodamine channels taken at 0.3 μm intervals were obtained using 1 s exposure times for each channel. For each strain time course, 500–800 cells were imaged over >5 time-course runs and >5 experiments.

Image stacks were deconvoluted, then analyzed and quantified for inner (internalized) vs. plasma membrane (PM) localized α-factor$^{Rhd}$ as ROIs using Fiji (*Schindelin et al., 2012*) as outlined (*Figure 5— figure supplement 1C*) using the scripts in supplemental materials (*Source code 1*). Ratio data of PM/ Total fluorescent intensity for each cell and each strain from multiple experiments were extracted to Excel, combined and re-grouped by time points then normalized to the zero timepoint. *Source data 2* were plotted and fit to non-linear fit one phase decay in Prism (GraphPad Software, San Diego, CA). One-way ANOVA statistical tests were performed for data at individual timepoints. Calculation of cell surface vs internal fluorescent α-factor ratios were determined by an observer blinded to the sample identity.

## Recombinant protein purification and crystallization

ANTH domain proteins and GST fusion proteins were expressed in *E. coli* BL21(DE3) strain upon induction with 0.5 M IPTG at 18 °C for 20 hr in LB media supplemented with ampicillin. Cells were lysed by French Press or OneShot cell disruptor (Constant Systems) in 287 mM NaCl, 2.7 mM KCl, 12 mM NaPO$_4$ buffer pH 7.4 and protease inhibitors (Complete EDTA-free, Roche). 6xHis tagged proteins were purified using Talon Co$^{2+}$ affinity resin (Takara Bio, USA) and eluted with 150 mM Imidazole in the lysis buffer pH 7.8. GST fusion proteins were purified over Glutathione (GSH) beads (GE Healthcare Biosciences, Pittsburg, PA) in PBS buffer supplemented with protease inhibitors and eluted with 20 mM GSH in PBS pH 7.4. Ub was purified as described previously (*Pashkova et al., 2010*).

For crystallization, CALM ANTH domain and Ub proteins were further purified on a HiLoad 16/60 Superdex 200 column (GE Healthcare Biosciences, Pittsburg, PA) equilibrated with 20 mM HEPES-NaOH, 150 mM NaCl buffer pH 7.3. Crystallization samples were obtained by combining two proteins CALM ANTH (33 mg/ml) and Ub (21 mg/ml) at molar ratio 1:1 at 20 °C. Co-crystals were grown by the hanging drop vapor diffusion method at 4 °C mixing equal amount of the CALM ANTH:Ub protein complex and reservoir solution containing 0.2 M NaCl, 20 % (w/v) Polyethylene glycol monomethyl ether 2,000 and 0.1 M MES monohydrate pH = 6. In 2 weeks, thick rod-shaped crystals appeared and were flash-cooled in liquid nitrogen directly from the crystallization solution. X-ray diffraction data of the crystal mounted frozen in a nitrogen gas stream at 100°K were collected to 2.4 Å on an 'in house' Rigaku RU-H3R X-ray machine with RAXIS-IV++ image plate detectors (University of Iowa, Protein Crystallography facility). Data were processed using XDS (*Kabsch, 2010*) and AIMLESS (*Evans, 2011*). Processing statistics are summarized in *Table 1*. Structure of CALM ANTH in complex with Ub was solved by molecular replacement with PHASER (*McCoy et al., 2007*) using published CALM structure PDB:1HF8 and Ub structure PDB:1UBQ as a starting model. Phenix.autobuild and phenix.refine were used to build and refine the model. Coot and PyMOL (The PyMOL Molecular Graphics System, Version 2.0 Schrödinger, LLC) were used to manually fit the structure and generate the structural figures. Software used in this project was curated by SBGrid (*Morin et al., 2013*).

## NMR binding experiments

$^{15}$N/$^1$H HSQC spectra of $^{15}$N-HIP1 ANTH with or without addition of native Ub, and $^{15}$N-Ub in presence of binding ANTH proteins or alone were collected at 25 °C on a Bruker Avance II 800 MHz spectrometer. Protein samples were prepared in 50 mM NaCl, 40 mM NaPO4, pH 6.95 (NMR buffer). Data were analyzed with SPARKY (T. D. Goddard and D. G. Kneller, SPARKY 3, UCSF, CA) and NMRView (One Moon Scientific, Westfield, NJ). Chemical shift perturbations were measured by comparing peak positions to $^{15}$N-Ub alone using $((0.2\Delta N^2 + \Delta H^2)^{1/2})$ to map residues in and near the interface Ub uses to bind ANTH domains. binding affinities were obtained by using the peak intensities of Ile30 of $^{15}$N-Ub in $^{15}$N/$^1$H HSQC spectra collected in the presence of various concentrations of the Ub-binding protein ANTH domain (WT or mutant). The bound fraction was calculated by measuring the difference in the peak intensity in the absence (free form) and presence (bound form) of the Ub-binding protein, then

divided by the peak intensity of the free form. These data were then fitted to a standard quadratic equation using GraphPad Prism (GraphPad Software) as reported previously (*Briggs et al., 2016*). The standard deviation from data fitting is reported. Every binding affinity experiment shown is a separate titration curve using the indicated ratios of proteins. To measure Paramagnetic Relaxation Enhancement (PRE) effects, mutant ANTH domain proteins containing a single cysteine at different positions were created. The Talon resin purified proteins were incubated with 1 mM MTSL in the elution buffer 18 hrs at 4 °C, then dialyzed into NMR buffer. For each spin-labeled Cys mutant protein sample, two HSQC spectra were collected with addition of $^{15}$N-Ub at 30 μM for two NMR samples. One sample had MTSL in the oxidized state, whereas second sample contained MTSL reduced by 5 mM sodium ascorbate as described (*Pashkova et al., 2013*). The degree of PRE effects was calculated as ratio of a particular residue peak intensity in the oxidized vs. reduced protein sample.

## Yeast strain construction

To generate yeast strains with genome-integrated versions of ANTH domain proteins containing Ub-binding site mutations, these following genes were cloned into pRS306 (*Sikorski and Hieter, 1989*), cut as listed below and sequentially recombined into yeast cells to URA+: For YAP1801 F213A, E214S: p6305 was cut with AleI. For YAP1802 F207A, E208A: p6306 was cut with PacI. For Sla2 L212A, D218A: p6307 was cut with XhoI. To obtain yeast that removed out *URA3* by spontaneous recombination (loop out), cells were plated on SD media containing 5-Fluoroorotic Acid.

To alter *STE2*, a KanMX DNA fragment from pUG6 (*Güldener et al., 1996*) and *STE2* wild type or a mutant *ste2* DNA fragment were amplified and recombined into a strain, colonies selected on YPD + G418 and checked for negative growth on SD-Leu.

To make yeast strains with integrated Ent1 and Ent2 UIM mutants, wild type Ent1 and Ent2 from yeast genomic DNA and the mutant UIM region from gene synthesized DNA were PCR amplified and assembled into pRS306 linearized with NotI/XhoI. For *ent1$^{ΔUIM}$*, p6712 was linearized with SpeI; for *ent2$^{ΔUIM}$* p6713 with NotI. Ura + transformants were then selected for Ura- loopout colonies by growth on SD-5FOA.

For yeast with integrated Ede1$^{ΔUBA}$ (ΔK$_{1342}$-stop), *URA3*_LoxP cassette was PCR amplified from pUG72 for recombination into the *EDE1* locus at the 3' end of the coding region. Colonies were selected on SD-Ura-Met plates. Recombination of the LoxP_*URA3*_LoxP site was induced with pPL5608_TEF1*-Cre_TRP1 (*MacDonald and Piper, 2015*). To make pGAD with human HIP1 and human HIP1R, corresponding ANTH domain was PCR amplified from human ORFeome library (Transomic Technologies) and assembled into p5808 cut (*Peterson et al., 2018*).

## Protein binding assays

Purified HA- or 6xHis-tagged recombinant proteins were incubated 1 hr at 20 °C with 0.1 mg of GST or GST fusion proteins bound to GSH beads (50 μl per sample) in PBS buffer in presence of 0.01 % Triton X100 and 0.1 mg BSA and casein, as described in *Pashkova et al., 2013*. After spin-wash, protein complexes were eluted from the beads with 40 mM GSH in PBS pH 7.4, resolved on SDS-PAGE then subjected to Western blotting analysis. Raw data available in *Source data 1*.

## Yeast two-hybrid test

Yeast MAT A (strain Y3581) expressing Gal4-BD diUb fusion protein from p5643 was grown on SD-Trp agar media. MAT α Y5725 yeast expressing Gal4-AD ANTH domain fusion proteins (wild type or mutants) from p6968, p6969, p7026, p7027 were grown on SD-Leu agar media. After mating on YPD plate, diploids were grown on SD-Leu-Trp at 28 °C and tested for ability to grow in absence of His and in the presence of 3-amino-triazole in the growth media (*Pashkova et al., 2016*). Raw data available in *Source data 1*.

## Acknowledgements

We wish to thank the contributions of the IIHG genomic sequencing core, and the CCOM NMR and protein crystallography cores. We also thank Dr. Tabitha Peterson for excellent editing of this manuscript. This research was supported, in whole or in part, by National Institutes of Health R01 GM58202 (to RCP), the American Diabetes Association post-doctoral fellowship (AYM) and by subsidy of core

facilities by the Carver College of Medicine. We dedicate this study to the memory and work of Linton Traub, who inspired this work.

# Additional information

## Funding

| Funder | Grant reference number | Author |
|---|---|---|
| National Institutes of Health | GM058202 | Robert C Piper |
| American Diabetes Association | Post-doctoral fellowship | Annabel Y Minard |

The funders had no role in study design, data collection and interpretation, or the decision to submit the work for publication.

## Author contributions

Natalya Pashkova, Conceptualization, Formal analysis, Methodology, Resources, Writing – original draft, Writing – review and editing; Lokesh Gakhar, Conceptualization, Formal analysis, Investigation, Methodology, Resources, Validation, Writing – original draft, Writing – review and editing; Liping Yu, Conceptualization, Formal analysis, Investigation, Methodology, Validation, Writing – review and editing; Nicholas J Schnicker, Conceptualization, Formal analysis, Methodology, Validation, Writing – review and editing; Annabel Y Minard, Software, Formal analysis, Funding acquisition; Stanley Winistorfer, Data curation, Methodology, Resources, Software, Writing – review and editing; Ivan E Johnson, Data curation, Formal analysis, Methodology, Resources, Software; Robert C Piper, Conceptualization, Data curation, Funding acquisition, Project administration, Supervision, Writing – original draft, Writing – review and editing

## Author ORCIDs

Nicholas J Schnicker (iD) http://orcid.org/0000-0002-5189-4943
Robert C Piper (iD) http://orcid.org/0000-0001-9995-5699

## Decision letter and Author response

Decision letter https://doi.org/10.7554/eLife.72583.sa1
Author response https://doi.org/10.7554/eLife.72583.sa2

# Additional files

## Supplementary files
- Transparent reporting form
- Supplementary file 1. Plasmids used.
- Supplementary file 2. Yeast strains used.
- Source data 1. Gel Scans and Catalog.
 Raw images of gels and yeast plates along with a catalog of the images included.
- Source data 2. Internalization Time-course Data used for *Figure 5*.
- Source code 1. Scripts for quantifying microscopy data.

## Data availability
Diffraction data have been deposited in PDB under the accession code 7JXV.

The following dataset was generated:

| Author(s) | Year | Dataset title | Dataset URL | Database and Identifier |
|---|---|---|---|---|
| Pashkova N, Gakhar L, Schnicker NJ, Piper RC | 2021 | ANTH domain of CALM (clathrin-assembly lymphoid myeloid leukemia protein) bound to ubiquitin | https://www.rcsb.org/structure/7JXV | RCSB Protein Data Bank, 7JXV |

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
