## [Editor Report]

This manuscript describes application of x-ray crystallography and NMR spectroscopy to define how ANTH domains, which are present in endocytic machinery, bind to ubiquitin. The structural study is well-done, and results extended to the domain family by using well designed, structure-based amino acid substitutions. The Ub-binding surface was found to overlap with that used to bind VAMP8 but the authors identify mutations that retain this interaction while losing binding to Ub. These mutations are then used in yeast-based functional assays to provide new insights into Ub-dependent internalization. This study is an impressive tour de force.

---

## [Decision Letter]

**Decision letter after peer review:**

Thank you for submitting your article "ANTH domains within CALM, HIP1R, and Sla2 recognize ubiquitin internalization signals" for consideration by *eLife*. Your article has been reviewed by 2 peer reviewers, and the evaluation has been overseen by a Reviewing Editor and Cynthia Wolberger as the Senior Editor. The following individual involved in review of your submission has agreed to reveal their identity: Florian Wilfling (Reviewer #2).

Essential revisions:

Overall, the reviewers feel that the paper is important and interesting. The reviewer's feel that 2 points need additional attention.

(1.) The authors do not provide data on how much the additional effect of the ANTH domain on Ub-binding compared to previously described Ub-domains in Ede1, Ent1 and Ent2 is. Internalization of fluorescence α-factor by the Ub-dependent STE2-∆C in Figure 5E should be performed and compared to ede1∆UBA ent1∆Ub ent2∆Ub to estimate the additional effect of the 3xANTH∆Ub combination. This seems critical to the conclusions of the paper.

(2.) The PDF version of the submitted paper had multiple issues related to missing or out of order figures, and figures that were not described appropriately. These are detailed by the comments of reviewer 1. Careful consideration of this will be important.

The reviewer comments below also contain a number of issues that can be addressed by changes to the text or figures but that do not require additional experiments. We look forward to seeing the revised version.

*Reviewer #1:*

This manuscript describes application of x-ray crystallography and NMR spectroscopy to define how ANTH domains, which are present in endocytic machinery, bind to ubiquitin. The structural study is well-done, and results extended to the domain family by using well designed, structure-based amino acid substitutions. The Ub-binding surface was found to overlap with that used to bind VAMP8 but the authors identify mutations that retain this interaction while losing binding to Ub. These mutations are then used in yeast-based functional assays to provide new insights into Ub-dependent internalization. This study is an impressive tour de force.

Some suggested textual changes are made below to make the manuscript more accessible to a general audience and easier to follow.

Figure 1D – the amino acids for which data is not available (such as Prolines) should be indicated as such on the surface diagrams and also in Supplementary Figure 1A.

The reference for Figure 2C at the bottom of p. 5 (pdf file) is missing. Including amino acid #s would also be helpful.

Issues with figure referencing made this manuscript challenging to follow. The figures are out of order, some panels not referenced, Figure S4F for example is not labeled as such but rather part of S4E, I can't find Figure S4-2. Figures are pre-referenced before the experimental data are described. p. 8-9 has quite a bit of introductory text that breaks the flow and distracts from the Results/experimental description with Figure 5C referenced along with Figure 5A (top of p. 9), before description of the experiment. Isn't Figure 5C the analysis of Figure 5D? It's not referenced where it is described in the text, directly following Figure 5D.

The supplementary figures that include the data points for Figure 5C, 5E, 5F should be referenced along with these figures at first appearance/experimental description in the text.

*Reviewer #2:*

In N. Pashkova et al., the authors follow up on results of a large-scale Y2H screen which revealed that the N-terminal portion of the endocytosis protein HIP1 binds ubiquitin (Ub). In the first part of the manuscript the authors clearly establish that the ANTH domain can serve as a Ub binding domain. Through biochemical analysis using GST-pulldown assays the authors showed that the ANTH domain alone was sufficient to bind Ub. Purification of different ANTH domains from various proteins and monitoring of their binding to Ub by HSQC NMR confirmed the initial findings and revealed equilibrium binding constants in the low µM range, which is in line with previous observed Ub binding domains (K.A. Swanson et al., JMB 2006).

To get insights into the mode of binding between the ANTH domain and Ub the authors pursued a crystal structure of CALM ANTH domain in complex with Ub. This revealed a CALM-ANTH:Ub interface. The key residues establishing the interface were seen in both the NMR experiments as well as the crystal structure. Paramagnetic Relaxation Enhancement (PRE) experiments confirmed the structural information and showed that Ub binds to the same region in solution. Based on this structural information the authors created mutants disrupting the binding between ANTH domains and Ub.

In the second part of the manuscript N. Pashkova et al., aim to investigate the role of the ANTH:Ub interaction on receptor internalization during endocytosis. For this they use two mutant variants of the α-factor receptor Ste2 which either is dependent or independent on Ub as an internalization signal as well as a mutant lacking both signals. With this system they test the effect of Ub-binding deficient ANTH proteins in Ste2 internalization. Using fluorescence microscopy, they measure indirectly the α-factor internalization by monitoring the fluorescence α-factor signal at the plasma membrane in chase experiments. From these experiments they conclude that Ub-dependent internalization is dependent on all Ub-binding domains within the endocytic machinery, including the Ub-binding property of the ANTH domains. The conclusion of the second part of this manuscript is mostly well supported by data, but some aspects would need to be clarified and extended.

(1.) The authors do not provide data on how much the additional effect of the ANTH domain on Ub-binding compared to previously described Ub-domains in Ede1, Ent1 and Ent2 is. Internalization of fluorescence α-factor by the Ub-dependent STE2-∆C in Figure 5E should be performed and compared to ede1∆UBA ent1∆Ub ent2∆Ub to estimate the additional effect of the 3xANTH∆Ub combination.

(2.) Figure 5D should contain microscopy images of all four Ste2 constructs at 10 min to allow comparison, since this timepoint is mainly used for analysis.

In summary, the manuscript by N. Pashkova et al., provides compelling evidence that the list of Ub-binding motifs with the endocytic machinery is much larger as currently anticipated. They provide initial evidence, that these new motifs might play a role in cargo internalization but more work is necessary to elucidate their mode of action. Additionally, it will be of great interest to see the contribution of those domains on assembly of the endocytic machinery.

The manuscript by N. Pashkova et al., seems to have a pdf compilation problem. Figure S4-2 is missing and many other figures are duplicated such as Figure 1. This needs to be fixed. Therefore, the bioRxiv version of the manuscript was partially used to evaluate the missing figures. Moreover, some figures such as Figure 3A and 3B are not annotated in the text.

In Figure 4B the amino acid labelling is too small to read this should be rearranged to make it readable.

Overall, the result that the ANTH domain serves as a Ub-binding domain is an intriguing finding. It would be very interesting to see if Ub-binding is important for assembly of the CME machinery. The authors provide a discussion but some experimental evidence would be very helpful and increase the impact of the work. For example, does patch life time of endocytic machinery proteins vary in the 3xANTH mutants or Ub∆ mutants?

The authors mention that 3xANTH∆Ub ede1∆UBA ent1∆Ub ent2∆Ub mutant does not show the same internalization defect as the Ste2-∆C, K>R. The SH3 domain of Sla1 another endocytic machinery protein has also been suggested to bind to ubiquitin (PMID: 17244534). Adding this mutation to the 3xANTH∆Ub ede1∆UBA ent1∆Ub ent2∆Ub could completely abolish receptor uptake and lead to a similar effect as seen in the Ste2-∆C, K>R. This would add additional value since it would show that most likely no additional Ub-binding mutants are necessary.

The authors mention data about pHIuorin fused Ste2 but do not provide any data. These data should be provided or the section should be removed from the manuscript.

---

## [Author Response]

Essential revisions:Overall, the reviewers feel that the paper is important and interesting. The reviewer’s feel that 2 points need additional attention.1) The authors do not provide data on how much the additional effect of the ANTH domain on Ub-binding compared to previously described Ub-domains in Ede1, Ent1 and Ent2 is. Internalization of fluorescence α-factor by the Ub-dependent STE2-∆C in Figure 5E should be performed and compared to ede1∆UBA ent1∆Ub ent2∆Ub to estimate the additional effect of the 3xANTH∆Ub combination. This seems critical to the conclusions of the paper.

The reviewers pointed out that we needed an additional control. This was to compare the effect of the 3xANTH∆Ub combination of mutations in the presence and absence of the ede1∆UBA ent1∆Ub ent2∆Ub combination of mutants. We now provide those data in Figure 5E and in supplemental data. They agree with previous published findings that no significant defect in Ste2 internalization is observed in either ede1∆UBA

ent1∆Ub ent2∆Ub, ede1∆UBA, or ent1∆Ub ent2∆Ub strains. Therefore, we are confident that ANTH domains do contribute to the recognition of ubiquitin as an internalization signal and do so in collaboration with other Ub-adaptors.

2) The PDF version of the submitted paper had multiple issues related to missing or out of order figures, and figures that were not described appropriately. These are detailed by the comments of reviewer 1. Careful consideration of this will be important.

We deeply apologize for the formatting issues of the paper. We struggled with the submission website. We have taken steps to ensure all figures are present and in the correct order. We do see places where we unfortunately confused readers and worked to clarify these throughout the manuscript.

Reviewer #1:This manuscript describes application of x-ray crystallography and NMR spectroscopy to define how ANTH domains, which are present in endocytic machinery, bind to ubiquitin. The structural study is well-done, and results extended to the domain family by using well designed, structure-based amino acid substitutions. The Ub-binding surface was found to overlap with that used to bind VAMP8 but the authors identify mutations that retain this interaction while losing binding to Ub. These mutations are then used in yeast-based functional assays to provide new insights into Ub-dependent internalization. This study is an impressive tour de force.Some suggested textual changes are made below to make the manuscript more accessible to a general audience and easier to follow.Figure 1D – the amino acids for which data is not available (such as Prolines) should be indicated as such on the surface diagrams and also in Supplementary Figure 1A.The reference for Figure 2C at the bottom of p. 5 (pdf file) is missing. Including amino acid #s would also be helpful.Issues with figure referencing made this manuscript challenging to follow. The figures are out of order, some panels not referenced, Figure S4F for example is not abelled as such but rather part of S4E, I can’t find Figure S4-2. Figures are pre-referenced before the experimental data are described. P. 8-9 has quite a bit of introductory text that breaks the flow and distracts from the Results/experimental description with Figure 5C referenced along with Figure 5A (top of p. 9), before description of the experiment. Isn’t Figure 5C the analysis of Figure 5D? It’s not referenced where it is described in the text, directly following Figure 5D.The supplementary figures that include the data points for Figure 5C, 5E, 5F should be referenced along with these figures at first appearance/experimental description in the text.

– The introductory text in the Results section leading to Figure 5 has been shortened, with part moved to the discussion to provide this context to the readers.

– Prolines of Ub in the ^1^H/^15^N HSQC data are indeed absent. We now note this in the figure legend for Figure 1 noting that only large observable chemical shift perturbations of observable backbone amides are mapped. We now point out in the graphs in S.Figure 1 which residue positions are not observed and explain this in the figure legend.

– Thank-you for the comments pointing out organizational flaws in the manuscript. We have better referenced the figures in the text and re-arranged Figure 5 to better explain the experimental analysis.

– The text now references Figure 2C and the residues highlighted are listed in the figure legend. We have also referenced figures such as S4 and S5 better. We have also reorganized Figure 5 as well since the micrographs shown next to a quantitated uptake graph are different experiments.

Reviewer #2:In N. Pashkova et al., the authors follow up on results of a large-scale Y2H screen which revealed that the N-terminal portion of the endocytosis protein HIP1 binds ubiquitin (Ub). In the first part of the manuscript the authors clearly establish that the ANTH domain can serve as a Ub binding domain. Through biochemical analysis using GST-pulldown assays the authors showed that the ANTH domain alone was sufficient to bind Ub. Purification of different ANTH domains from various proteins and monitoring of their binding to Ub by HSQC NMR confirmed the initial findings and revealed equilibrium binding constants in the low µM range, which is in line with previous observed Ub binding domains (K.A. Swanson et al., JMB 2006).To get insights into the mode of binding between the ANTH domain and Ub the authors pursued a crystal structure of CALM ANTH domain in complex with Ub. This revealed a CALM-ANTH:Ub interface. The key residues establishing the interface were seen in both the NMR experiments as well as the crystal structure. Paramagnetic Relaxation Enhancement (PRE) experiments confirmed the structural information and showed that Ub binds to the same region in solution. Based on this structural information the authors created mutants disrupting the binding between ANTH domains and Ub.In the second part of the manuscript N. Pashkova et al., aim to investigate the role of the ANTH:Ub interaction on receptor internalization during endocytosis. For this they use two mutant variants of the α-factor receptor Ste2 which either is dependent or independent on Ub as an internalization signal as well as a mutant lacking both signals. With this system they test the effect of Ub-binding deficient ANTH proteins in Ste2 internalization. Using fluorescence microscopy, they measure indirectly the α-factor internalization by monitoring the fluorescence α-factor signal at the plasma membrane in chase experiments. From these experiments they conclude that Ub-dependent internalization is dependent on all Ub-binding domains within the endocytic machinery, including the Ub-binding property of the ANTH domains. The conclusion of the second part of this manuscript is mostly well supported by data, but some aspects would need to be clarified and extended.(1.) The authors do not provide data on how much the additional effect of the ANTH domain on Ub-binding compared to previously described Ub-domains in Ede1, Ent1 and Ent2 is. Internalization of fluorescence α-factor by the Ub-dependent STE2-∆C in Figure 5E should be performed and compared to ede1∆UBA ent1∆Ub ent2∆Ub to estimate the additional effect of the 3xANTH∆Ub combination.(2.) Figure 5D should contain microscopy images of all four Ste2 constructs at 10 min to allow comparison, since this timepoint is mainly used for analysis.In summary, the manuscript by N. Pashkova et al., provides compelling evidence that the list of Ub-binding motifs with the endocytic machinery is much larger as currently anticipated. They provide initial evidence, that these new motifs might play a role in cargo internalization but more work is necessary to elucidate their mode of action. Additionally, it will be of great interest to see the contribution of those domains on assembly of the endocytic machinery.The manuscript by N. Pashkova et al., seems to have a pdf compilation problem. Figure S4-2 is missing and many other figures are duplicated such as Figure 1. This needs to be fixed. Therefore, the bioRxiv version of the manuscript was partially used to evaluate the missing figures. Moreover, some figures such as Figure 3A and 3B are not annotated in the text.In Figure 4B the amino acid labelling is too small to read this should be rearranged to make it readable.Overall, the result that the ANTH domain serves as a Ub-binding domain is an intriguing finding. It would be very interesting to see if Ub-binding is important for assembly of the CME machinery. The authors provide a discussion but some experimental evidence would be very helpful and increase the impact of the work. For example, does patch life time of endocytic machinery proteins vary in the 3xANTH mutants or Ub∆ mutants?The authors mention that 3xANTH∆Ub ede1∆UBA ent1∆Ub ent2∆Ub mutant does not show the same internalization defect as the Ste2-∆C, K>R. The SH3 domain of Sla1 another endocytic machinery protein has also been suggested to bind to ubiquitin (PMID: 17244534). Adding this mutation to the 3xANTH∆Ub ede1∆UBA ent1∆Ub ent2∆Ub could completely abolish receptor uptake and lead to a similar effect as seen in the Ste2-∆C, K>R. This would add additional value since it would show that most likely no additional Ub-binding mutants are necessary.The authors mention data about pHIuorin fused Ste2 but do not provide any data. These data should be provided or the section should be removed from the manuscript.

– As suggested, we now provide data on internalization of Ste2∆C in ede1∆UBA ent1∆Ub ent2∆Ub mutants with and without additional mutation of the Ub-binding surfaces in the 3 yeast ANTH domain proteins. We agree this is an important addition to the work and helps support the idea that Ub-bindng by ANTH domains contributes to Ub-dependent internalization.

– We now include pictures of yeast expressing the 4 different Ste2 proteins used in the study as they internalize α factor. Shown are the time 0 timepoint while they are at 0°C and a timepoint 10 min after warming. This now substitutes for the earlier micrographs shown 0min and 20 min after warming. We see that the arrangement of panels in Figure 5 were confusing in that the quantitation of how these 4 different Ste2 proteins should have been juxtaposed to micrograph panels to the right. We have re-arranged Figure 5 to first show what a time course looks like in which we image different cells at different timepoints to avoid quenching artifacts. Then we show quantitation of all the different types of Ste2 proteins we use to keep these concepts separate. The requested micrographs of T0 and T10 are provided in SFig5B and show the same cells imaged at different times to allow readers to appreciate the kinetics of internalization.

– Again apologies for having to go to the BioRvix version of the paper to get the proper PDF file. We really appreciate your effort. Figure S4-2 is in this version. Figures3A and 3B are specifically called out

– The amino acid lettering in Figure 4B is indeed way too small to be of use and its also too small in SFig2. We have taken that out and simply labelled it based on amino acid position. In response to a concern to R1, we have also designated proline residues explicitly as well, which do not give an HSQC backbone spectra.

– The idea of whether Ub-binding alters the assembly kinetics of the CME apparatus is indeed really interesting and something we discuss in the paper. To do this right, we would want to couple that to also looking at cargo and see if assembly around Ub-dependent cargo is different from Ub-independent cargo in the midst of thee mutations. We feel this is an important but quite technically challenging question, we only know of one paper that follows cargo into assembling yeast CME puncta and the regimen to synchronize that movement is fairly different from what we use here. Without solving this, we think the cursory answer we would obtain would be unsatisfying and distracting and possibly a deterrent for others that are currently better suited to address this.

– Our discussion does indeed mention the possibility that Ub-binding by SlaI could be the one, or one of many remaining Ub adaptors for CME. This speaks to a larger effort to explain all of Ub-dependent internalization activity in yeast vs what we wanted to focus on in this manuscript which was whether Ubbinding by ANTH domains contributed to Ub-dependent internalization. Answering the larger question of how all CME Ub-binding proteins contribute represents making and analyzing a number of complicated strains that we could not complete. For instance, we have had problems altering SLA1. And as a curiosity, we pick up tons of mutations in AVL9 in these complex strains when we alter EDE1, which prevents Ste2 trafficking to the plasma membrane. All that is potentially really interesting, but outside the scope of this focused study on whether Ub-binding by ANTH domains does something.

– In terms of the effects of pHluorin, we have removed that from the main text and made a technical note in the methods section and show data in supplementary figures that prompted us to switch to un-tagged Ste2. We point out that it still internalizes and that pHluorin itself does not appear to provide an internalization signal since Ste2 without its SlaI binding site (∆C) and lacking its normal ubiquitinated lysines (K>R) does not internalize α factor when appended to pHluorin.